Microbiology **Spectrum**

# Blocking the shikimate pathway amplifies the impact of carvacrol on biofilm formation in *Candida albicans*

Ali Molaeitabari,[1] Tanya E. S. Dahms[1]

**ABSTRACT**  *Candida albicans* typically thrives in a commensal relationship with humans but is also an opportunistic fungal pathogen. As an opportunistic pathogen, *C. albicans* relies heavily on its ability to assimilate nutrients, for which it must compete with the host and other microorganisms. Amino acid biosynthesis, sensing, and uptake play pivotal roles in *C. albicans* growth and pathogenicity. *C. albicans* biosynthesizes aromatic amino acids and co-enzyme Q *de novo* through the shikimate pathway, including the Aro1, Aro2, and Aro7 enzymes, but also has amino acid transporters for uptake from the environment. Thus, antifungal approaches targeting aromatic amino acid biosynthesis must simultaneously inhibit amino acid biosynthesis and uptake. Herein, we investigate the plant-based antifungal, carvacrol, in conjunction with aromatic amino acid biosynthetic mutants, as a potential anti-candidal strategy. Growth of the WT, *ARO2*, and *ARO7* strains were inhibited by 150 µg/mL carvacrol, whereas the *ARO1* mutant was slightly more sensitive (with MIC 125 µg/mL). All repressed mutants exposed to carvacrol are partially rescued in the presence of para-aminobenzoic acid (PABA) (CoQ precursor), indicating that blocking the shikimate pathway impacts both aromatic amino acid and CoQ biosynthesis. Moreover, carvacrol at sublethal concentrations significantly inhibits *ARO1* adhesion and hyphal formation, along with pre-attached and pre-formed hyphae, ultimately impacting biofilm metabolic activity and biomass accumulation and significantly reducing biofilm growth. In summary, carvacrol increases the sensitivity of *C. albicans* to *ARO1* repression, with attenuated adhesion, hyphal formation, mycelial growth and biofilm formation, likely by blocking aromatic amino acid uptake.

**IMPORTANCE**  The opportunistic pathogen *Candida albicans* remains the leading cause of candidemia and invasive candidiasis (IC), causing significant morbidity and mortality in immunocompromised patients. Our current arsenal of effective antifungal drugs is limited in number, mechanistic diversity, and efficacy, are cytotoxic and associated with antifungal resistance, necessitating the development of novel antifungals and combination therapies. Here, we show how simultaneously blocking the shikimate pathway, through *ARO1* repression, and disrupting aromatic amino acid uptake by carvacrol prevent *C. albicans* biofilm formation. Thus, inhibitors of the Aro1 enzyme in combination with carvacrol are expected to shut down *C. albicans* biofilm formation and virulence.

**KEYWORDS**  *ARO1*, aromatic amino acids, biofilm, carvacrol, *Candida albicans*, co-enzyme Q, biofilm, PABA, yeast-hyphal transition, shikimate pathway

**Peer Reviewer** Alireza Khodavandi, Islamic Azad University, Gachsaran, Iran

Address correspondence to Tanya E. S. Dahms, tanya.dahms@uregina.ca.

The authors declare no conflict of interest.

See the funding table on p. 16.

Worldwide, approximately one billion people are suffering from fungal diseases, of which 150 million are life-threatening infections that cause more than 1.5 million deaths annually (1, 2). *Candida* spp. are still the most common cause of invasive fungal infections, and nearly 700,000 cases of invasive candidiasis (IC) are estimated annually (1, 3). Invasive candidiasis and candidemia are caused primarily by five species: *C. albicans*, *C. glabrata*, *C. tropicalis*, *C. parapsilosis*, and *C. krusei*, accounting for 90%–92% of all

cases (3–5). Among these species, *Candida albicans* is the leading cause, accounting for approximately 82% of all cases (6).

*C. albicans* is a commensal microorganism of the human microflora found in the gastrointestinal (GI) tract, skin, mouth, and vaginal microbiota (5, 7, 8), but it is also an opportunistic pathogen. In relatively healthy individuals, *C. albicans* gives rise to superficial (skin and mucosa) infections, whereas in immunocompromised patients, *C. albicans* can bring about invasive and deep-seated infections (9, 10). There are a number of factors that promote infection by *C. albicans*, including biofilm formation (9–11).

Biofilms are the complex three-dimensional microbial structure that enhance *C. albicans* virulence through resistance to antifungal drugs, protection against host immune factors, transfer of genetic material, cell signaling, and enhanced adhesion (9, 12, 13). Biofilm formation is a cyclic process that begins with adhesion of planktonic *C. albicans* (yeast form) to biotic (e.g., host cell) or abiotic (e.g., catheter) surfaces, then proliferation as a microcolony of yeast, germ tubes, pseudohyphae, and hyphae, followed by maturation as a complex three-dimensional structure develops, including the secretion and accumulation of protective extracellular matrix (ECM), and finally planktonic dispersal to other niches (9, 13–15).

Nutrient metabolism is not only essential for cell growth and survival but also *C. albicans* pathogenesis and virulence (16–19). Eukaryotic fungi have biosynthetic pathways for all aromatic amino acids, whereas human cells are only capable of converting L-Phe to L-Tyr (20, 21), making *de novo* aromatic amino acid biosynthesis a potential target for new antifungals (22). The shikimate pathway (Fig. 1) is the linear portion of the biosynthetic cascade that uses carbohydrate precursors, phosphoenolpyruvate (PEP), and erythrose 4-phosphate (E4P) for the *de novo* synthesis of aromatic amino acids, ubiquinone/co-enzyme Q (CoQ), and folic acid (vitamin B9) (20, 22–24). The pathway, involving the Aro (AROmatic amino acid requiring) proteins, begins with Aro3/Aro4, followed by Aro1, a multi-enzyme complex, and finally Aro2 that produces chorismate, the precursor for the synthesis of aromatic amino acids and ubiquinone, whereas Aro7 produces the precursor of L-Tyr and L-Phe (Fig. 1) (20, 22, 24, 25). Aromatic amino acids are crucial for protein synthesis and function, whereas CoQ localizes to the plasma membrane, several endomembrane systems, and most importantly the inner mitochondrial membrane as part of the electron transport chain (ETC) (20, 22–24, 26). Yeast can use para-aminobenzoic acid (PABA), derived from chorismate, as a precursor for CoQ (25).

Global analysis of the *C. albicans* gene replacement and conditional expression (GRACE) shows *ARO1*, *ARO2*, and *ARO7* genes to be essential (27, 28). The *C. albicans ARO1* knockdown has altered cell wall composition and architecture and attenuated virulence in a host infection model that is rescued by supplementing with aromatic amino acids (29, 30), along with altered adhesion and biofilm formation through the altered gene expression of agglutinin-like sequences 1 (*ALS1*), *ALS3*, and extent of cell elongation 1 (*ECE1*) (21, 31–33). Therefore, *ARO1* is an attractive target for antifungal drug development.

Amino acids, as key nitrogen sources, can also be sensed and taken up by permeases (34) that play key roles in *C. albicans* growth, survival, and virulence (35, 36). At least three permeases participate in aromatic amino acid uptake through proton-driven permease symport, with general amino acid permease 2 (*Ca*Gap2) and *Ca*Gap6 transporting aromatic amino acids, and *Ca*Gap1 transporting only Phe (29, 37). All are suspected aromatic amino acid sensors, and activators of the rat sarcoma virus–cyclic adenosine monophosphate–protein kinase A (Ras-cAMP-PKA) pathway, which is important for hyphal morphogenesis and biofilm formation (37). Thus, antifungal compounds capable of impacting amino acid transport are expected to be synergistic with those that shut down amino acid biosynthesis.

Ideal antifungal drugs with mechanistic diversity, efficacy, coupled with low cytotoxicity and side effects, and minimal antifungal resistance are limited, necessitating the development of novel antifungals (38–42). There is a resurged interest in natural

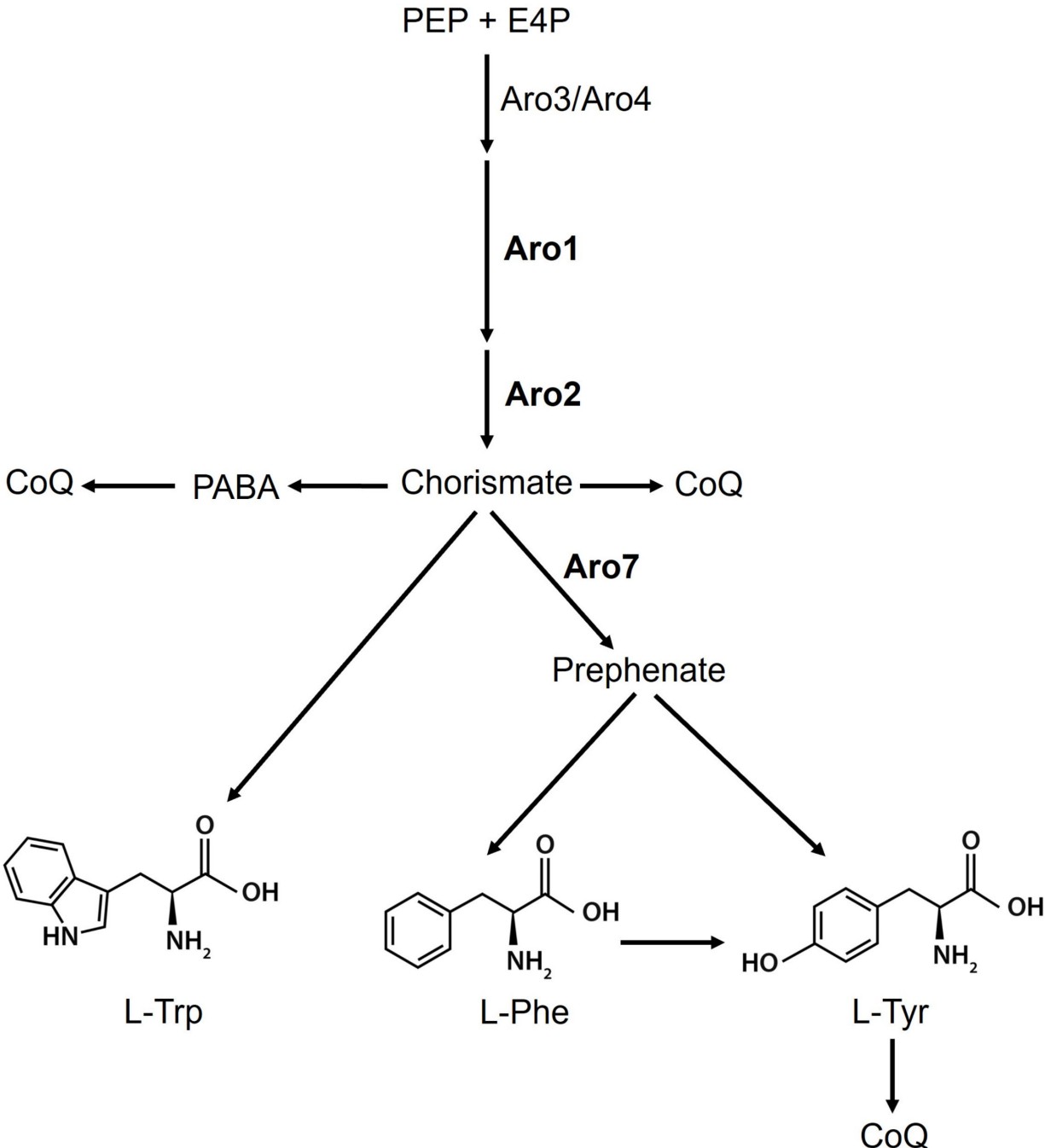

**FIG 1** *C. albicans* shikimate pathway and its relationship with other metabolic pathways. The shikimate pathway converts erythrose-4-phosphate and phosphoenolpyruvate to chorismate in seven enzymatic steps. Chorismate is a precursor to various aromatic compounds, such as the aromatic amino acids and their derivatives, as well as co-enzyme Q (CoQ). The linear pathway begins with Aro3/Aro4 (3-deoxy-D-arabino-heptulonate-7-phosphate (DAHP) synthase), followed by Aro1, a multi-enzyme complex catalyzing reactions 2 to 6 (20), and finally Aro2 (chorismate synthase) that produces chorismate, the branch point precursor for the synthesis of aromatic amino acids, folate, and CoQ. Aro7 (chorismate mutase) produces the precursor for both L-Tyr and L-Phe. Adapted from References (22, 24, 25). Created in https://BioRender.com.

products, including plant-based essential oils, as antifungals (43). Carvacrol, a phenolic monoterpenoid, exhibits low minimum inhibitory concentrations (MICs) against fungi and has a broad spectrum of antifungal activity (44–46). Moreover, carvacrol has low cytotoxicity and is effective in the treatment of systemic candidiasis with *C. albicans* and non-*C. albicans* (*C. krusei* and *tropicalis*) in mouse model, making it a promising potential

natural antifungal against candidal infections for clinical use (30, 47, 48). This compound impairs the cell membrane and endoplasmic reticulum (ER) by disrupting ergosterol biosynthesis (44, 49). Since the fungal ER is the site of ergosterol (50) and cell wall component biosynthesis, the impact of carvacrol may be an indirect consequence of ER stress (51). However, as the most abundant fungal sterol, ergosterol plays a critical role in composition, localization, and activity of some plasma membrane proteins, with the potential to impact amino acid permeases (50, 52, 53). We recently reported that carvacrol disrupts vacuolar membrane integrity, ultimately inhibiting hyphal progression (54). Haploid deletion chemical–genetic profiling in *Saccharomyces cerevisiae* showed that deletion of tryptophan biosynthetic genes (e.g., *TRP1*, *TRP2*, *TRP3*, *TRP4*, *ARO1*, and *ARO2*) causes hypersensitivity to carvacrol (51). Moreover, the phenolic compound eugenol interferes with aromatic amino acid uptake in *S. cerevisiae* likely by binding to the active sites of membrane-bound permeases (tyrosine amino acid transporter 1 [Tat1] and Gap1) (55). Since carvacrol is also a phenolic compound, with structural similarity to the side chains of Tyr and Phe, it has the potential to competitively block their transport. Here, we study the impact of carvacrol on aromatic amino acid uptake using *C. albicans* aromatic amino acid biosynthetic shut off strains and how this affects virulence factors. For the first time, we show how simultaneously blocking aromatic amino acid biosynthesis by *ARO1* repression and carvacrol treatment impact *C. albicans* adhesion, hyphal and biofilm formation.

## MATERIALS AND METHODS

### Antifungal

Carvacrol (natural, 99%) was purchased from Sigma-Aldrich (SKU# W224511, St. Louis, MO, USA). Dimethyl sulfoxide (DMSO, 99.6+%, Aldrich Chemical Co., Milwaukee, WI, USA) at a final concentration of 1% in various media was used to solubilize carvacrol and had no impact on cell growth.

### Microbial strains and growth conditions

*C. albicans* SC5314 (wild-type [WT] strain) along with the *C. albicans ARO1*, *ARO2*, and *ARO7* conditional **G**ene **R**eplacement **A**nd **C**onditional **E**xpression (GRACE) mutants (*orfX::his3::hisG/his3::hisG leu2::tetR-GAL4AD-URA3/LEU2*) were kind gifts of Dr. Malcolm Whiteway (Department of Biology, Concordia University, Montreal, CA). Strains stored as 50% glycerol stocks at −80°C were revived on yeast peptone dextrose (YPD) agar plates containing 1% yeast extract (Sigma-Aldrich, St. Louis, MO, USA), 2% bacto-peptone (Difco, BD Biosciences, NJ, USA), 2% D-glucose (Sigma-Aldrich, St. Louis, MO, USA), 2% agar (VWR)), and stored at 4°C for 21 days. For preparation of overnight cultures, single colonies were routinely inoculated into YPD broth and incubated at 30°C with shaking (200 rpm) for 11–14 h prior to each experiment.

Liquid media were supplemented with 2 and 20 µg/mL doxycycline (DOX) and solid media with 0.05 and 0.2 µg/mL DOX, which were sufficient to turn down (partially suppress) or shut off (fully suppressed) the conditional knock-out genes, respectively. All media were supplemented with 50 µg/mL uridine (56).

For MIC assays, an overnight culture at logarithmic phase was adjusted to an optical density at 600 nm ($OD_{600}$) of 0.001 (equivalent to $1.2 \times 10^5$ cells/mL or 0.5 McFarland standard) by diluting in YPD broth. Overnight cultures (9–11 h) at mid-logarithmic phase ($OD_{600}$ ~0.6–0.9) having more metabolically active cells were monitored at 30 min intervals using a microplate reader (BioTek; synergy HTX multi-mode reader; Winooski, VT, USA) for 24 h at 30°C for mutants and the WT strain in YPD broth (Fig. S1). The overnight cultures were centrifuged and then resuspended into 10 mL of fresh SC or RPMI media with/without DOX and grown at 30°C with shaking (200 rpm; 3 h), and then diluted (to ~$10^7$ CFU/mL) in appropriate media to generate experimental cultures for any given biochemical assay. For pre-treated cells, phosphate-buffered saline (PBS; 1 L Mili

Q water with one pouch of powdered salt [Sigma-Aldrich, St. Louis, MO, USA]; 10 mM phosphate, 138 mM NaCl, 27 mM KCl, pH 7.4), was used to remove carvacrol, eliminate any carryover effect, and normalize to $10^5$ cells/mL (optimal, unsaturated signal).

## Gene essentiality verification and screening

The gene essentiality assay was used to confirm cell viability, as previously described (27, 57), with slight modification. Briefly, all overnight cultures were serially diluted ($10^5$, $10^4$, $10^3$, $10^2$, and $10^1$ cells/mL), and 5 µL of each aliquot was spotted on synthetic defined (SD) minimal solid medium (0.17% yeast nitrogen base [YNB] without amino acids [BioShop Canada Inc.]) with 0.5% $NH_4SO_4$ (Fluka), 2% dextrose, and 2% agar plates supplemented with DOX or not. The plates were incubated at 30°C and photographed for 3 d.

## Minimum inhibitory concentration (MIC) assay

The MIC of carvacrol was determined using the twofold broth serial microdilution assay for all strains following the guidelines of the Clinical and Laboratory Standards Institute (CLSI) and previously reported methods (58) with slight modifications. Briefly, 100 µL of carvacrol (600 and 1,000 µg/mL) in 1% DMSO/YPD was serially diluted in triplicate in the wells of flat-bottom polystyrene 96-well microtiter plates (Sarstedt, Nümbrecht, Germany) to which experimental cultures were added. Growth controls consisted of untreated *C. albicans* in media, and blanks contained only carvacrol in 1% DMSO/YPD. The 96-well plates were incubated with shaking (30°C, 200 rpm, 24 h), and the $OD_{600}$ was measured on a microtiter plate reader (Biotek Epoch; Northern Vermont, USA) to determine the minimum concentration of carvacrol, leading to 100% growth inhibition compared with growth controls. Unless otherwise stated, all assays were performed in triplicate for each of three biological replicates.

## Auxotrophic supplement spot assay

Auxotrophic supplement was used to determine the impact of PABA on repressed mutants following carvacrol exposure. Synthetic complete (SC) solid medium (0.62% YNB without PABA; MPBiomedical, Solon, Ohio, USA) was supplemented with 0.17% synthetic amino acids, excluding aromatic amino acids (USBiological Life Sciences, Salem, MA, USA), 0.5% $NH_4SO_4$ (Fluka), and 2% agar with/without 0.25 mM PABA (Sigma-Aldrich, St. Louis, MO, USA). Overnight cultures were serially diluted ($10^5$, $10^4$, $10^3$, and $10^2$ cells/mL), and 5 µL of each aliquot was spotted on SC solid medium plates with/without PABA in the presence and absence of DOX and carvacrol. Following incubation at 30°C, the plates were photographed for 3 d.

## Adhesion assay

Adhesion to a polystyrene cell culture microplate surface was assessed for *C. albicans* WT and repressed and non-repressed mutants pre-treated with carvacrol, as previously described (59), with slight modification. Briefly, cells from an overnight YPD culture were washed and then sub-cultured into fresh SC synthetic complete (SC) medium (0.17% YNB; BioShop Canada Inc.), both of which were supplemented with 0.17% synthetic non-aromatic amino acids (USBiological Life Sciences, Salem, MA, USA), 0.5% $NH_4SO_4$ (Fluka), and 2 mM aromatic amino acids. Following incubation (30°C; 200 rpm, 3 h) ±DOX, cells were washed and resuspended ($1 \times 10^7$ CFU/mL) in SC broth medium ±DOX. Next, all strains were pre-treated with or without (negative control) carvacrol at 1/2 MIC and MIC in SC broth medium ±DOX in the wells of a flat-bottom polystyrene 96-well microtiter plate (Sarstedt, Nümbrecht, Germany) and incubated (30°C, 200 rpm, 4 h). The resulting cells were washed with PBS three times to remove carvacrol and to eliminate any carryover, and approximately $1 \times 10^5$ cells/mL were suspended in 500 mL YPD per well in 24-well tissue culture plates (Cat # 83.3922, Sarstedt, Nümbrecht, Germany) and incubated (1.5 h, 37°C, 5% $CO_2$) under static conditions. The plate was carefully washed three times with sterile PBS to remove non-adherent cells, the adherent cells were

scraped from each well and resuspended in 500 µL PBS, and the $OD_{600}$ was evaluated (Biotek Epoch; Northern Vermont, USA).

The adhesion assay was also used to evaluate the impact of carvacrol on pre-attached WT, repressed, and non-repressed mutant strains to the surface of polystyrene cell culture microplates. Briefly, approximately $1 \times 10^7$ CFU/mL of mid-logarithmic phase WT and mutant strains (ARO1, ARO2, ARO7) was washed and resuspended in YPD broth medium, and 500 µL aliquots were transferred to 24-well tissue culture plates (Cat # 83.3922, Sarstedt, Nümbrecht, Germany). Plates were incubated (1.5 h, 37°C, 5% $CO_2$) under static conditions and gently washed (PBS, 3×) to remove non-adherent cells. SC broth medium (500 mL) ±DOX and with or without carvacrol at 1/2 MIC and MIC were added to each well and incubated (37°C, 5% $CO_2$, 4 h). Non-adherent cells were removed, and adherent cells assessed as above.

## Serum-induced hyphal formation assay

The yeast to hyphal transition assay was used to assess the impact of carvacrol on *C. albicans* ARO1 according to published protocols (54, 60), with slight modification. Briefly, a yeast suspension ($1 \times 10^7$ CFU/mL) was prepared from sub-culture in pre-warmed SC with 10% FBS and deposited into a 24-well plate (Sardstedt, Nümbrecht, Germany) containing 500 µL of carvacrol serially diluted to 1/2 MIC and MIC ±DOX. Following shaking incubation (4 h, 37°C, 200 rpm), the cells were washed three times, resuspended in PBS, and stained with calcofluor white (CFW; 0.01 µg/mL; Fluka Analytical) to highlight the septa and cell wall. An aliquot (5 µL) of a treated or untreated *C. albicans* ARO1 suspension in PBS was pipetted onto sterile glass microscope slides, covered with a clean coverslip, and sealed with nail polish. Morphological changes (budding yeast, germ tube, and hyphae) were identified using the transmitted light configuration of an epifluorescence microscope ($\lambda_{ex}$ = 365 nm; $\lambda_{em}$ = 435 nm; Carl Zeiss Axio Observer Z1 inverted microscope, Oberkochen, Germany) at 63×. Results were evaluated based on the average number of germ-tube forming cells/100 cells from each biological replicate. The same method (54, 60), with slight modification, was used to assess the impact of carvacrol on pre-formed hyphae. Briefly, pre-formed hyphae were prepared by adding 10% FBS to an overnight culture (37°C), then suspended in pre-warmed SC with 10% FBS ($1 \times 10^7$ CFU/mL), deposited into a 24-well plate (Sardstedt, Nümbrecht, Germany) containing 500 µL of carvacrol in SC broth medium at 1/2 MIC or MIC ±DOX, and incubated (4 h, 37°C, 200 rpm). Following three washes with PBS, the cells were resuspended, stained with CFW and assessed as above.

## Mycelial growth assay

Mycelial growth inhibition was monitored in spider solid media (1% peptone, 1% yeast extract, 1% manitol, 0.5% NaCl, and 0.2% $K_2HPO_4$) for the *C. albicans* WT strain and ARO1, ARO2, and ARO7 mutants according to a published protocol (61), with slight modification. Briefly, after washing, overnight cultures were resuspended ($1 \times 10^7$ CFU/mL) in PBS, and 2 µL aliquots of *C. albicans* WT and mutant strains were spotted onto 12-well plates (CAT# 83.3921.500; Sarstedt, Nümbrecht, Germany) containing solid spider media ± DOX with or without carvacrol at 1/2 MIC and 1/4 MIC. The plates were incubated (6 days, 37 °C), and hyphal growth at the colony edges was imaged and captured on a digital camera using a stereomicroscope at 2× and 4× magnification.

Inhibition of mycelial growth was also assessed for *C. albicans* WT and mutant strains pre-treated with carvacrol according to the literature (61), with slight modification. Briefly, cells from an overnight culture were sub-cultured into fresh SC ±DOX and incubated (30°C, 200 rpm, 3 h) followed by washing and resuspension ($1 \times 10^7$ CFU/mL) with SC broth ±DOX. All strains either without (negative control) or pretreated with carvacrol at 1/2 MIC in SC broth medium ±DOX were incubated (30°C, 200 rpm, 4 h) in wells of flat-bottom polystyrene 96-well microtiter plates (Sarstedt, Nümbrecht, Germany). The cells were then washed three times and resuspended to $1 \times 10^5$ cells/mL in PBS, and 2 µL aliquots were spotted onto 12-well plates (CAT# 83.3921.500; Sarstedt,

Nümbrecht, Germany) containing solid spider media prepared with or without carvacrol ±DOX. The plates were incubated (6 days, 37 °C), and hyphal growth at the colony edges imaged as above.

## Biofilm assays

Biofilms were assessed by growth in Roswell Park Memorial Institute (RPMI) 1640 (Sigma-Aldrich, St. Louis, MO, USA) medium buffered with 4-morpholinepropanesulfonic acid (MOPS, 0.165 M, pH 7.0; Sigma-Aldrich, St. Louis, MO, USA) and with 10% FBS.

For biofilms of pre-treated cells, experimental cultures were treated with carvacrol at 1/2 MIC and MIC as described above, and *C. albicans* without carvacrol served as a negative control. Next, cells were washed with PBS three times, and their density was adjusted to $1 \times 10^5$ cells/mL. Aliquots (200 µL) were transferred to 96-well polystyrene tissue culture plates for static incubation (37°C, 1.5 h) during the initial adhesion phase. Non-adherent cells were removed with PBS, fresh RPMI-MOPS medium was added, and the plates incubated with shaking (48 h, 37°C, 75 rpm).

For pre-formed biofilm, experimental cultures were washed with PBS, resuspended in RPMI-MOPS medium, and 200 µL aliquots were transferred to 96-well polystyrene tissue culture plates. After 1.5 h static incubation at 37°C, non-adherent cells were removed by gentle washing with PBS and reconstituted in fresh RPMI-MOPS medium. To allow biofilm formation, plates were incubated with shaking (48 h, 37°C, 75 rpm) and after 48 h incubation, RPMI-MOPS medium ±DOX, without or with carvacrol at 1/2 MIC and MIC was added to each well.

Biofilms were gently washed once with 200 µL PBS, and biofilm formation was quantitatively assessed for pre-treated cells and pre-formed biofilm using 2,3-bis-(2-methoxy-4-nitro-5-sulfophenyl)-2H-tetrazolium-5-carboxanilide (XTT; Invitrogen) and crystal violet (CV; Sigma-Aldrich, St. Louis, MO, USA) assays according to previously published protocols (56).

## XTT assay

The XTT reduction assay examined the metabolic activity of the biofilms (56). Briefly, rinsed biofilms were incubated with 0.2 mg/mL of XTT in PBS and 0.004 mM menadione (Sigma-Aldrich) in absolute ethanol at 37°C for 2.5 h under static conditions in the dark. An aliquot (100 µL) of the supernatant was transferred to a new 96-well plate, and absorbance at 490 nm was recorded (BioTek; synergy HTX multi-mode reader; Winooski, VT, USA).

## Crystal violet (CV) assay

Biomass accumulation was assessed using crystal violet. Briefly, rinsed biofilms were air dried for 45 min, stained with 0.4% aqueous CV solution, and incubated at room temperature (RT) for 45 min, washed twice with sterile Milli Q water, and then de-stained with 95% ethanol at RT for 45 min. An aliquot (100 µL) of the de-stained solution was transferred to a new 96-well plate, and the absorbance at 495 nm was recorded (BioTek; synergy HTX multi-mode reader; Winooski, VT, USA).

Percent inhibition was calculated as follows:

% Inhibition = 100 − [(A EOC/A control) × 100], where A is absorbance at 495 nm

## Statistical analyses

GraphPad Prism (Version 9.0; La Jolla, CA, USA) was used to analyze the data by one-way analysis of variance (ANOVA) with Dunnett's multiple post-test to compare all data versus control for ungrouped data with more than two variables. Error bars represent the standard error of the mean (SEM) for all biological replicates. Statistical significance is denoted in figures by asterisks, * ($P < 0.05$), ** ($P < 0.01$), *** ($P < 0.001$), **** ($P < 0.0001$), and lack thereof (ns; $P > 0.05$).

## RESULTS

### The *ARO1* gene is essential for *C. albicans* viability

The growth phenotype was distinct for each of WT and *ARO1, ARO2*, and *ARO7* on SD solid medium ±DOX (Fig. S2). The conditionally repressed *ARO1* mutant did not grow in SD medium with DOX, confirming that the *ARO1* gene is essential for cell viability, while the repressed *ARO2* and *ARO7* mutants had strong and medium growth defects, respectively, on SD medium with DOX.

### The *ARO1* heterozygous mutant is more sensitive to carvacrol

The relative sensitivity of the WT, *ARO1, ARO2,* and *ARO7* mutant strains to carvacrol was assessed by MIC assay (Fig. 2). The planktonic growth of the WT strain and two heterozygous mutants (*ARO2* and *ARO7*) was effectively inhibited by 150 µg/mL carvacrol, whereas the *ARO1* heterozygous mutant was slightly more sensitive (125 µg/mL). Based on the 1/2 MIC values, *ARO1* is the most sensitive to carvacrol. Unless otherwise stated, carvacrol at 1/2 MIC and MIC were used for all subsequent experiments.

### Repressed *ARO1* is partially rescued with PABA

PABA (CoQ precursor) supplementation was used to assess the impact of shikimate pathway repression on CoQ production (Fig. S3). When repressed, *ARO1* did not grow, and *ARO2* growth was defective on SC solid medium, which could partially be rescued with PABA. Under these conditions, carvacrol had only a minor impact on growth (Fig. 3).

### Carvacrol pre- and post-treatment reduces adhesion of repressed *ARO1*

To determine the impact of carvacrol pre-treatment on cell adhesion, the conditionally repressed *ARO1*, *ARO2*, and *ARO7* mutants, non-repressed mutants, and WT strain were incubated on cell culture polystyrene microplates following exposure to sublethal (1/2

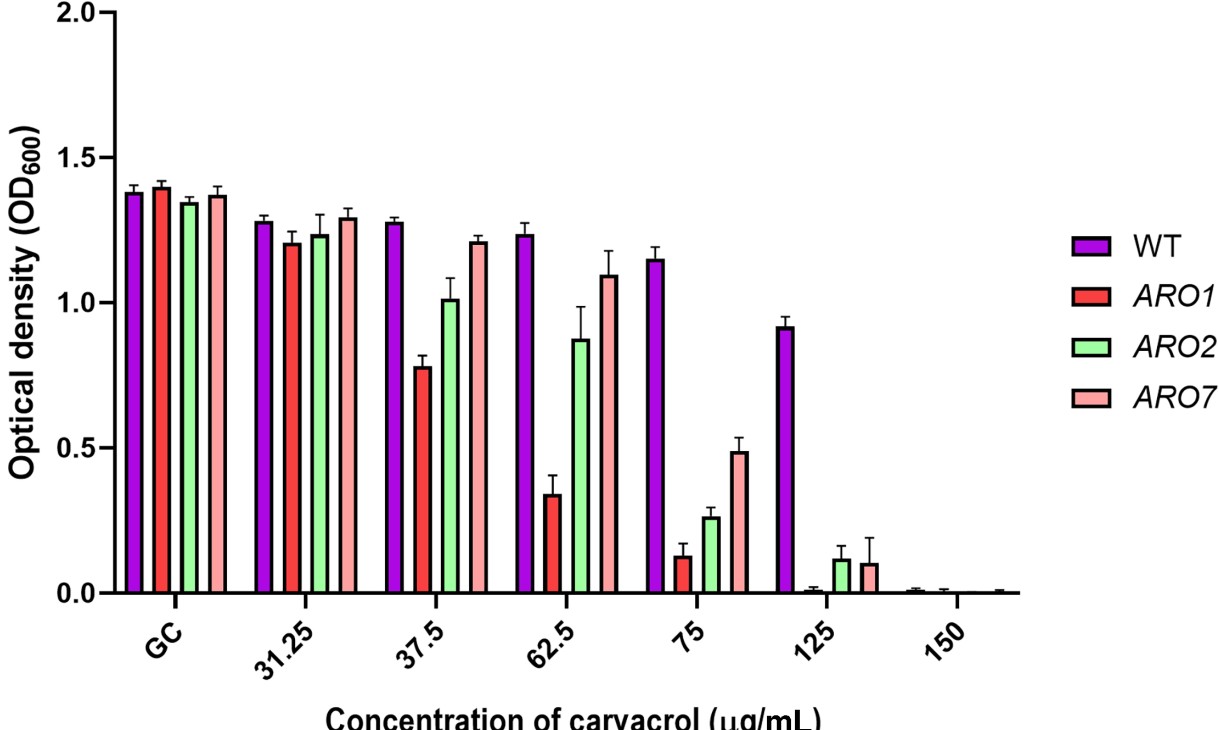

**FIG 2** The *ARO1* heterozygous mutant is more sensitive to carvacrol. Overnight cultures exposed to carvacrol were incubated for 24 h at 30°C, with shaking at 200 rpm. GC: growth control.

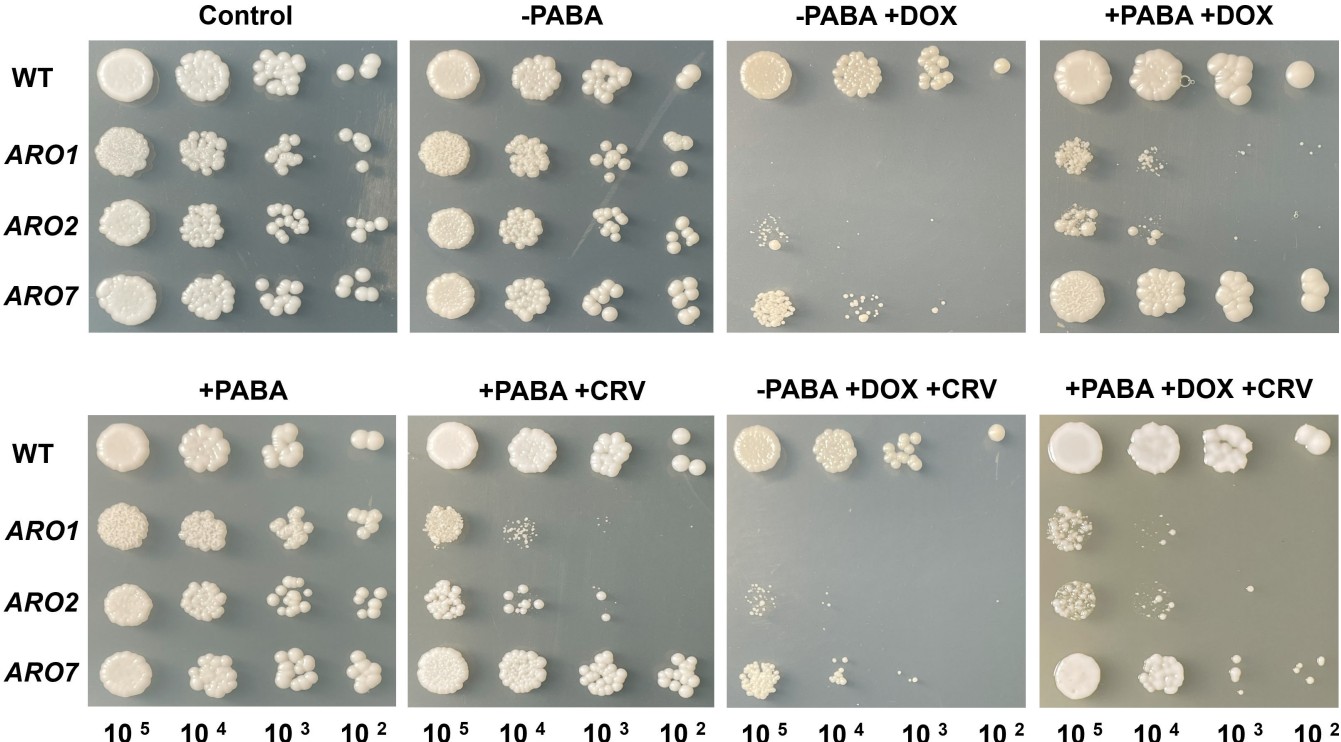

**FIG 3** Repressed *ARO1*, *ARO2*, and *ARO7* mutants treated with carvacrol are partially rescued by PABA (CoQ precursor). The WT strain and *ARO1*, *ARO2*, and *ARO7* mutant overnight cultures were 10-fold serially diluted ($10^5$, $10^4$, $10^3$, and $10^2$ CFU) and spotted on SC medium lacking aromatic amino acids supplemented with PABA, either in the presence or absence of DOX or carvacrol at 1/2 MIC. The plates were incubated at 30°C for 2 days and then photographed. CRV: carvacrol; PABA: para-aminobenzoic acid.

MIC) and lethal (MIC; positive control) levels of carvacrol. Repressed and non-repressed *ARO1* pre-treated with carvacrol at 1/2 MIC had adhesion to polystyrene were significantly weaker than those of the *ARO2* and *ARO7* mutants under the same conditions (Fig. 4a), with repressed *ARO1* being the weakest. Moreover, conditionally repressed *ARO1* adhesion following exposure to carvacrol at 1/2 MIC was twofold less than that of non-repressed *ARO1* under the same conditions (Fig. 4a).

To determine the impact of carvacrol on pre-attached cells, the WT and conditionally repressed and non-repressed *ARO1*, *ARO2*, and *ARO7* mutants were pre-attached to polystyrene surfaces and exposed to sublethal (1/2 MIC) and lethal (MIC) levels of carvacrol. The presence of carvacrol at MIC significantly reduced adhesion for all strains, but more so for the *ARO1* mutant (Fig. 4b). The adhesion of WT and non-repressed *ARO1*, *ARO2*, and *ARO7* mutants exposed to carvacrol at 1/2 MIC were nearly identical, whereas *ARO1* had significantly reduced adhesion compared with *ARO2* and *ARO7* mutants when repressed (Fig. 4b).

## Carvacrol inhibits mycelial growth in repressed *ARO1*

WT, repressed and non-repressed *ARO1*, *ARO2*, and *ARO7* strains grown on spider media containing carvacrol at 1/4 and 1/2 MIC produced fewer mycelia compared with controls (Fig. 5a and b). Untreated and treated WT colonies were round with regular wrinkles, whereas untreated and treated repressed and non-repressed mutant colonies had highly irregular edges (Fig. 5a and Fig. S4a). The non-repressed *ARO1* control had dense mycelial masses at their edges, resembling the WT strain and non-repressed *ARO2* and *ARO7* control colonies, but with slightly less prominent mycelia (Fig. 5a and b; Fig. S4a). Non-repressed *ARO1* colonies exposed to carvacrol at 1/4 and 1/2 MIC had less mycelial growth at their edges compared with controls, whereas repressed *ARO1* colonies under the same conditions had little to no mycelial growth (Fig. 5a and b), representing greater

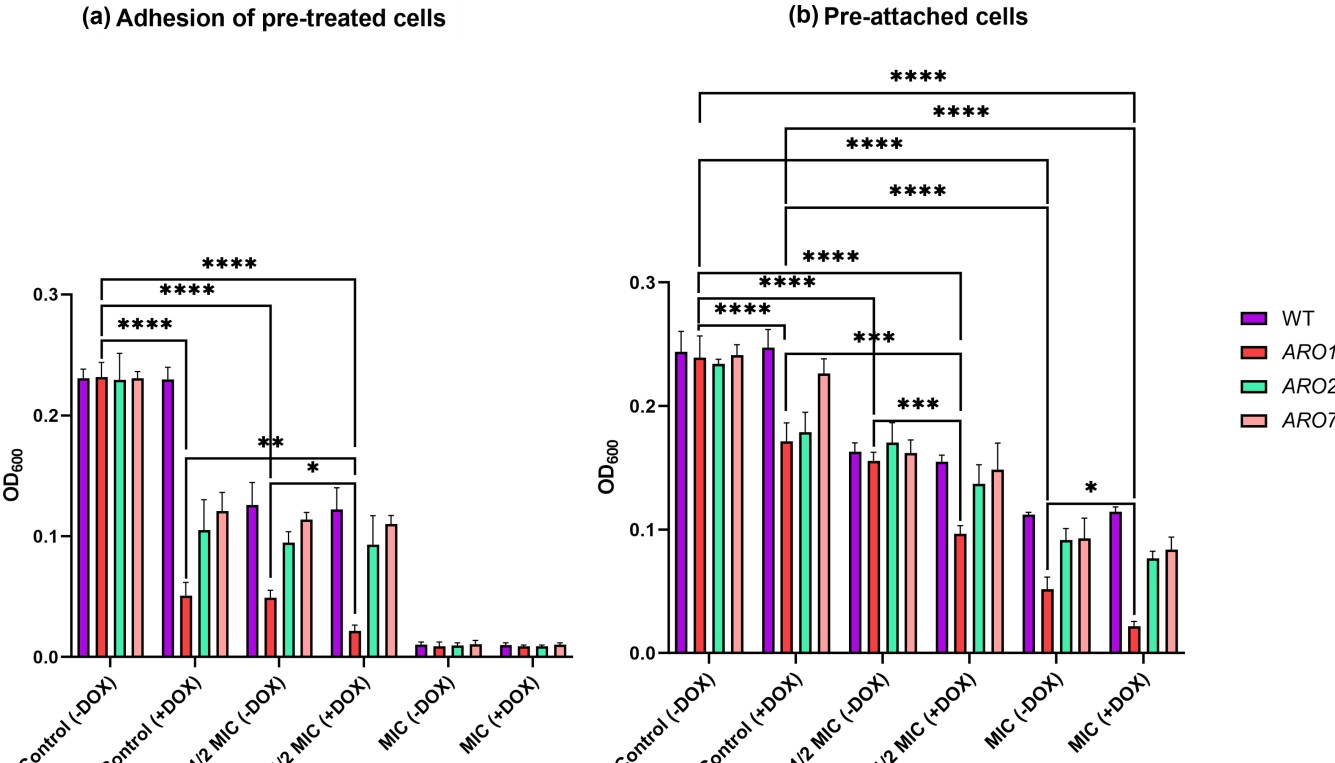

**FIG 4** Carvacrol reduces adhesion of pre-treated and pre-attached repressed mutant strains. Cells from overnight culture were sub-cultured in SC broth ±DOX for 3 h. (a) The WT strain and conditional mutants were pre-treated with carvacrol at 1/2 MIC and MIC (positive control) in SC liquid medium ±DOX for 4 h. Following washing and resuspension in YPD liquid medium, cells were incubated under static conditions for 1.5 h at 37°C, 5% $CO_2$ in a polystyrene microplate. (b) To test the impact of carvacrol on pre-adhered cells, the WT strain and conditional mutants in YPD broth medium were incubated in the same manner and exposed to carvacrol at 1/2 MIC and MIC in SC liquid medium ±DOX for 4 h at 37°C under static conditions. For both assays, adhered cells were washed three times to remove planktonic cells, scraped from the polystyrene surface, and adhesion evaluated by measuring the cell density ($OD_{600}$). Statistical significance from a one-way ANOVA is denoted by * ($P < 0.05$), ** ($P < 0.01$), *** ($P < 0.001$), **** ($P < 0.0001$).

inhibition of morphological switching in the repressed *ARO1* mutant exposed to carvacrol. Although mycelial growth in repressed and non-repressed *ARO2* exposed to carvacrol at 1/2 MIC resembled *ARO7* under the same conditions, at 1/4 MIC they differed (Fig. 5b; Fig. S4a). Moreover, the non-repressed and repressed *ARO1* mutants pre-treated for 4 h with carvacrol at 1/2 MIC, and further incubated for 6 days following carvacrol removal had reduced mycelial growth compared with the non-repressed and repressed *ARO2* and *ARO7* mutants (Fig. 5c and d; Fig. S4b). The non-repressed *ARO1* control had similarly dense mycelial mass as the WT strain both in the presence and absence of carvacrol, but the WT strain treated with carvacrol at 1/2 MIC and the non-repressed *ARO1* control had slightly less prominent mycelia (Fig. 5c).

## Carvacrol hinders hyphal formation and alters pre-formed hyphal morphology in repressed *ARO1*

Since the repressed *ARO1* mutant colony exposed to carvacrol at 1/2 MIC in spider media lacked mycelia, the impact of carvacrol on *ARO1* (repressed/non-repressed) hyphal formation and pre-formed hyphae was examined in 10% FBS serum. Carvacrol at 1/2 MIC and MIC (positive control) was a potent inhibitor of the yeast to hyphal transition in the non-repressed *ARO1* mutant (Fig. 6a and b), with a 51% reduction or complete inhibition, respectively, as compared with untreated controls. The repressed *ARO1* mutant exposed to carvacrol at 1/2 MIC had a 77% reduction in hyphae compared with untreated controls (Fig. 6a and b), showing a significant impact of carvacrol on the repressed *ARO1* mutant.

## (a) Mycelial growth inhibition

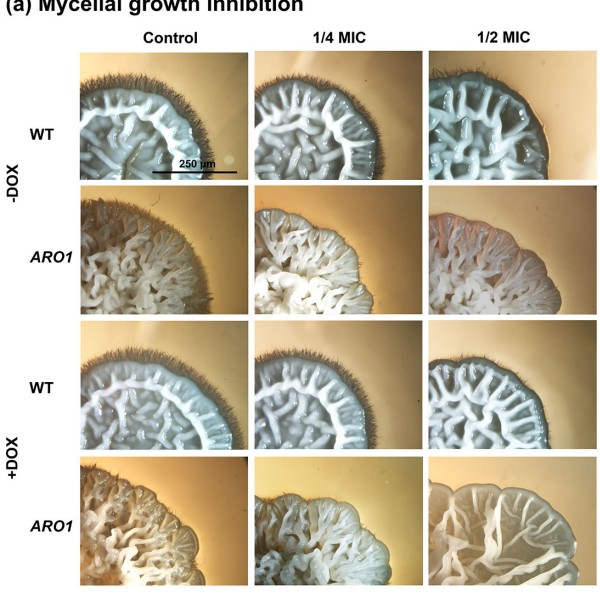

### (b)

| Distribution of mycelial mass under stereomicroscope | | | | |
|---|---|---|---|---|
| **Mycelial growth inhibition** | | | | |
| | | Control | 1/4 MIC | 1/2 MIC |
| **-DOX** | WT | ++++ | +++ | ++ |
| | *ARO1* | ++++ | ++ | + |
| | *ARO2* | ++++ | ++ | ++ |
| | *ARO7* | ++++ | +++ | ++ |
| **+DOX** | WT | ++++ | +++ | ++ |
| | *ARO1* | ++ | + | – |
| | *ARO2* | ++ | ++ | ++ |
| | *ARO7* | +++ | +++ | ++ |

## (c) Mycelial of pre-treated cells

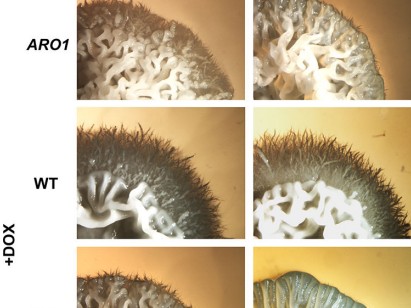

### (d)

| Distribution of mycelial mass under stereomicroscope | | |
|---|---|---|
| **Pre-treated mycelial cells** | | |
| | | Control | 1/2 MIC |
| **-DOX** | WT | ++++ | +++ |
| | *ARO1* | ++++ | ++ |
| | *ARO2* | ++++ | ++ |
| | *ARO7* | ++++ | +++ |
| **+DOX** | WT | ++++ | +++ |
| | *ARO1* | ++ | + |
| | *ARO2* | ++ | ++ |
| | *ARO7* | +++ | +++ |

**FIG 5** Carvacrol impacts *C. albicans* WT, *ARO1*, *ARO2*, and *ARO7* mycelial growth. Representative stereoscopic bright-field images of the WT and mutant strains on spider media agar plates show colony morphology with (a) constant exposure to carvacrol (6 days), (b) 4 h pre-treatment with carvacrol, followed by 6 day incubation in the absence of carvacrol. Results for WT, *ARO1*, *ARO2*, and *ARO7* strains are summarized in tabular format in panels (c) and (d), for which ++++ (maximal), +++, ++, and + (minimal) indicate the relative amount of mycelial growth. Scale bar = 250 µm, applicable to all images.

As might be expected, when the repressed *ARO1* mutant was exposed to carvacrol at 1/2 and full MIC (positive control), most cells appear as yeast and budding yeast (85% and 93%), with only 15% and 7% as hyphae and germ tubes, respectively (Fig. 6c and d). This is in comparison to the non-repressed *ARO1* mutant under the same conditions, for which 69% and 70% were yeast and budding yeast, and 31% and 30% were hyphae and germ tubes, respectively (Fig. 6c and d).

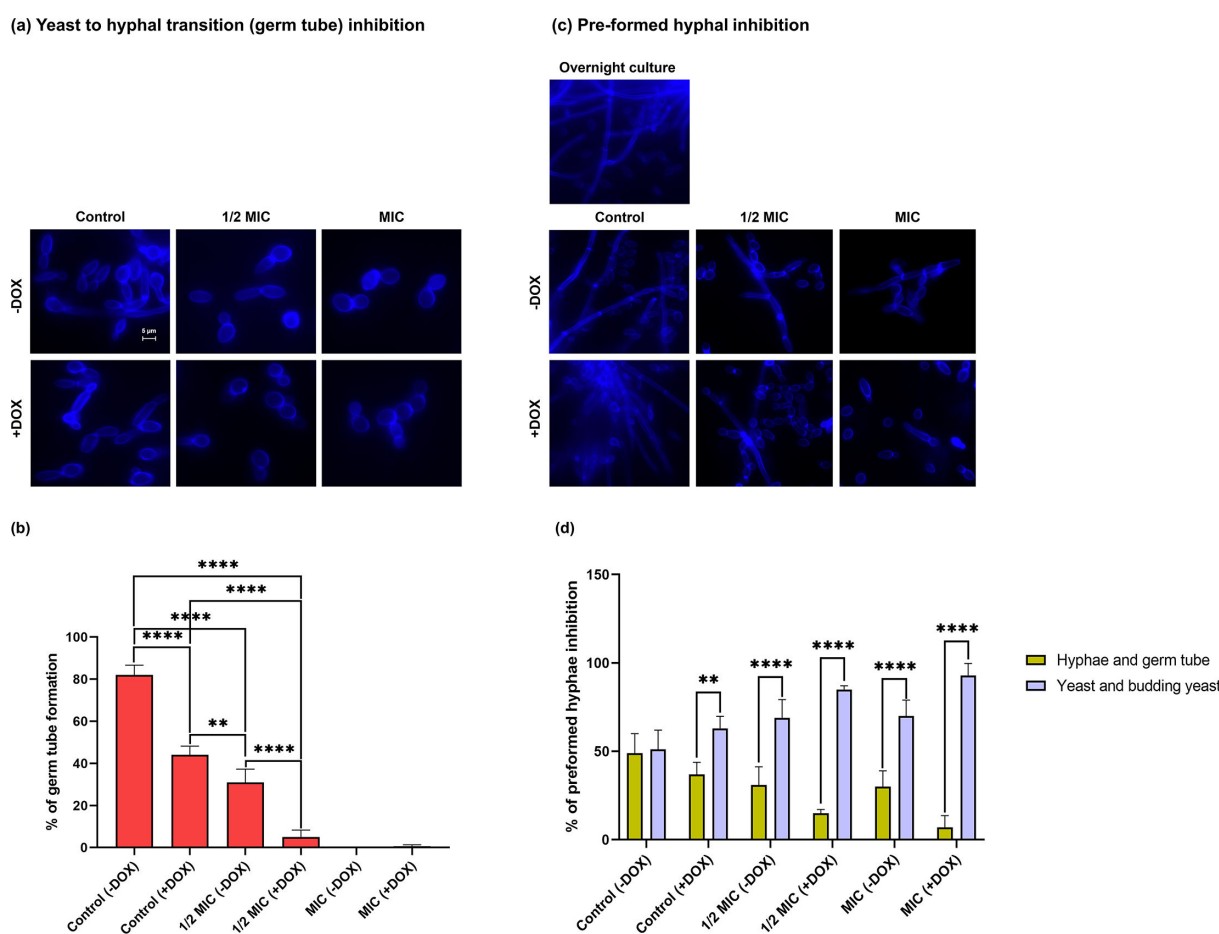

**FIG 6** Carvacrol impacts *C. albicans ARO1* hyphal formation. Representative epifluorescence microscopy ($\lambda_{ex}$ = 365 nm; $\lambda_{em}$ = 435 nm) images show the impact of 4 h carvacrol exposure at 1/2 MIC or MIC on *C. albicans ARO1* mutant a) hyphal formation and c) pre-formed hyphae when grown in SC medium with 10% FBS ±DOX. Scale bar for the control is 5 μm and applies to all images. Bar graphs show carvacrol c) inhibits the yeast to hyphal transition and most interestingly d) reverses the pre-formed hyphal morphology for the *ARO1* mutant. Statistical significance, determined by a one-way ANOVA, is denoted by ** ($P < 0.01$), **** ($P < 0.0001$).

## *ARO1* repression enhances the impact of carvacrol pre-treatment on biofilm growth

The impact of carvacrol on metabolic activity and biomass accumulation of biofilms formed from the *ARO1*, *ARO2*, and *ARO7* mutants was assessed by XTT and CV, respectively. The biofilm metabolic activity and biomass of the repressed *ARO1* mutant were 55% and 60%, respectively, that of non-repressed controls. Following exposure to carvacrol at MIC, biofilm was further reduced to 17% and 19% of the controls, respectively (Fig. 7). Taken together, the data indicate that the conditional repression of *ARO1* increases biofilm sensitivity to carvacrol.

The *ARO1* mutant biofilm, both repressed and non-repressed, has fewer metabolically compromised cells but a greater mass than the WT strain (Fig. 7b), consistent with previous observations (56), suggesting that the *ARO1* biofilm may produce more extracellular matrix or consist of larger cells.

## Carvacrol alters the properties of pre-formed biofilm

Pre-formed biofilms (48 h) of the WT and conditional mutants had significantly reduced biomass for all strains in the presence of carvacrol, but more so for the *ARO1* mutant (Fig. 8a).

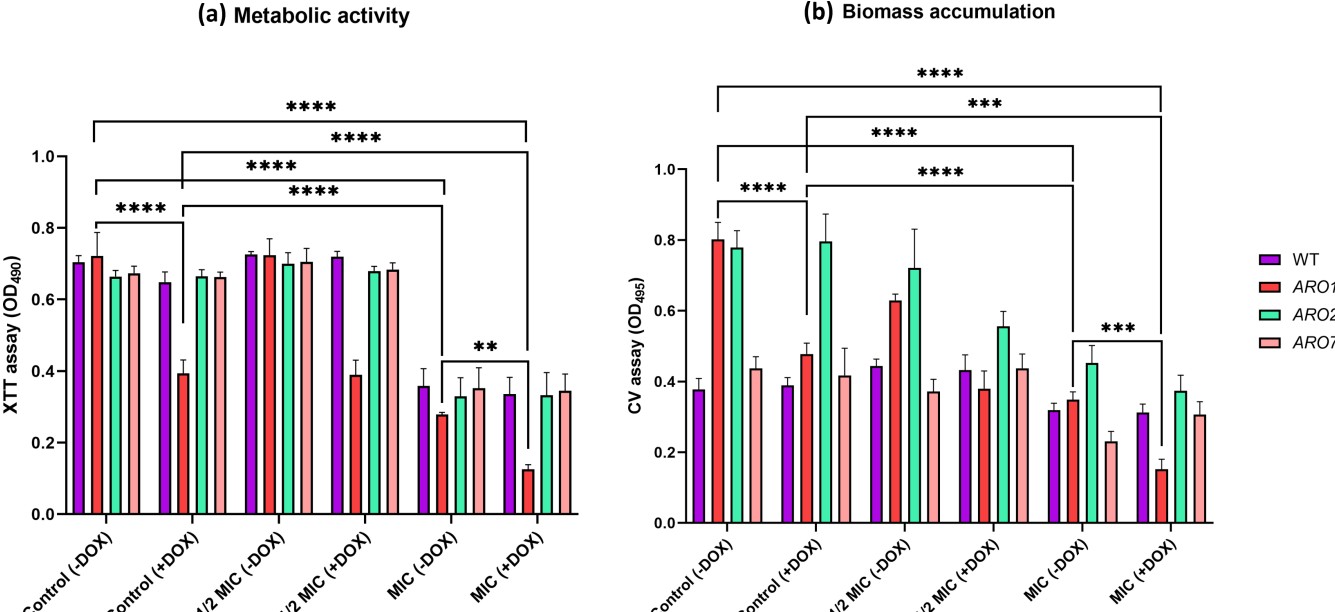

**FIG 7** Repression of the *ARO1* gene amplifies the impact of carvacrol pre-treatment on biofilm growth. Biofilms of pre-treated WT strain and conditional mutants were quantitatively evaluated using the (a) XTT (metabolic activity) and (b) CV (biomass accumulation) assays. Following exposure of the WT strain and conditional mutants to carvacrol at 1/2 MIC and MIC in YPD liquid medium ±DOX for 24 h at 30°C with 200 rpm shaking, carvacrol was removed, and biofilms were grown in RPMI-MOPS for 48 h at 37°C with shaking. Statistical significance, evaluated by a one-way ANOVA, is denoted by ** ($P < 0.01$), *** ($P < 0.001$), **** ($P < 0.0001$).

Carvacrol had no significant impact on the metabolic activity of *ARO2*, *ARO7* (repressed or not), and WT pre-formed biofilms but significantly reduced the metabolic activity of biofilms formed from repressed *ARO1* in comparison to that of both the WT strain and non-repressed *ARO1* (Fig. 8b). Since the RPMI medium supports the formation of a robust biofilm after 48 h, with the saturation threshold sometimes hiding subtle effects (62, 63), metabolic activity was assessed at 24 h for less well-developed biofilms. Metabolic activity of the non-repressed and repressed *ARO1* biofilms at 24 h was significantly reduced following carvacrol exposure (Fig. 8c).

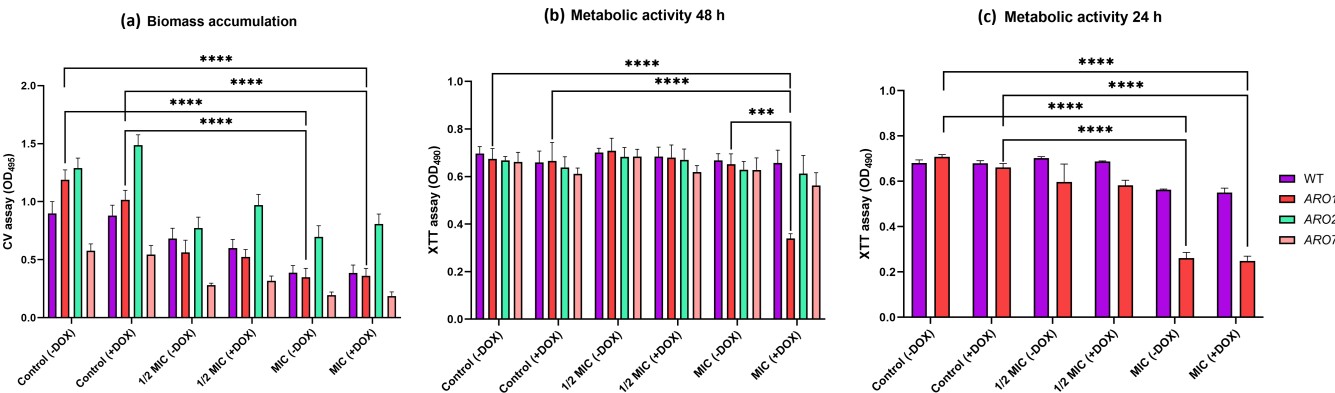

**FIG 8** Pre-formed *ARO1* mutant biofilms are inhibited by carvacrol. The response of pre-formed biofilms of WT and conditional mutants to carvacrol was quantitatively evaluated using the (a) CV assay (biomass accumulation) for 48 h biofilms, and the XTT assay (metabolic activity) for (b) 48 h and (c) 24 h biofilms. Biofilms were grown in RPMI-MOPS medium for 24 or 48 h at 37°C with shaking at 75 rpm, then treated with carvacrol at 1/2 MIC and MIC in RPMI-MOPS medium ±DOX for 24 h at 37°C with shaking at 75 rpm. Statistical significance, evaluated by a one-way ANOVA, is denoted by *** ($P < 0.001$), **** ($P < 0.0001$).

## DISCUSSION

*C. albicans* is an opportunistic fungal pathogen, which grows commensally with human hosts, and therefore must adapt to host niches to survive under different physiological conditions. The survival of *C. albicans* in/on the host depends on its capacity to assimilate nutrients in competition with the host and other microbiota (35). Since amino acids are used to synthesize proteins, converted to key metabolic intermediates, and used as both nitrogen and carbon sources, amino acid sensing and uptake play critical roles in *C. albicans* growth and pathogenicity (35, 36).

Of the three conditional mutants associated with the shikimate pathway and aromatic amino acid biosynthesis, only *ARO1* is essential for cell viability (Fig. S2), as expected (20, 21). Although carvacrol was lethal for both mutants and WT strains, the *ARO1* heterozygous mutant is more sensitive to carvacrol than *ARO2* and *ARO7* (Fig. 2). Since the repressed *ARO1* mutant is unable to synthesize aromatic amino acids *de novo* nor grow without aromatic amino acids (Fig. S2) (21), it serves to highlight the effect of carvacrol on aromatic amino acid sensing and uptake. Based on the structural similarity of carvacrol to the side chains of aromatic amino acids, such as Tyr and Phe, and our data in this study, we hypothesized that carvacrol can inhibit aromatic amino acid uptake (Fig. 9).

*C. albicans* cell adhesion, the first step in biofilm formation and hyphal formation, is critical for mucosal colonization and biofilm biogenesis (64, 65). Inhibition of biofilm formation at early stages in repressed and non-repressed *C. albicans ARO1* by carvacrol implies its impact on initial adhesion (Fig. 4). *ARO1* knockdown has been shown to affect adhesion through altered expression of cell adhesin genes *ALS1*, *ALS3*, and *HWP1* (21, 31), which are regulated by target of rapamycin (TOR) signaling (66) and known to be

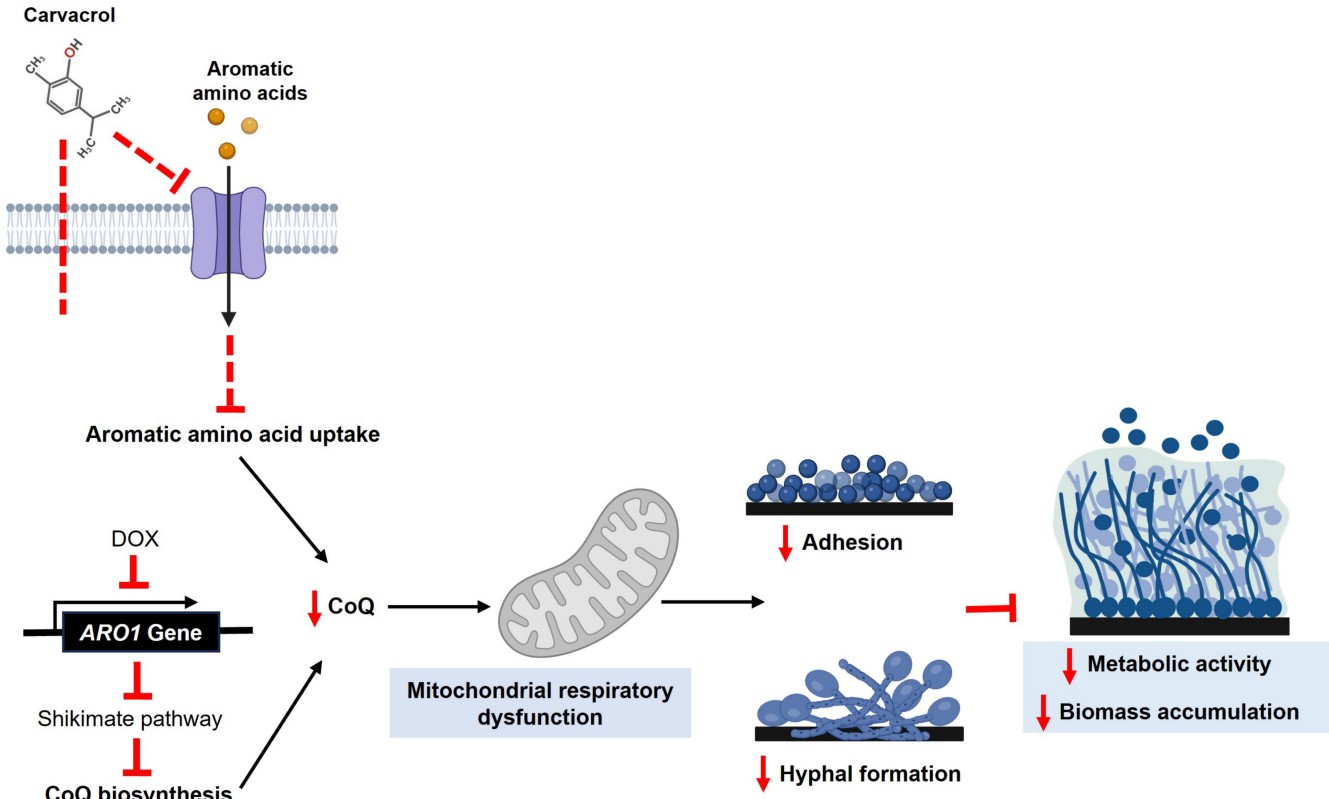

**FIG 9** Schematic of the impact of repressed *ARO1* with carvacrol exposure on biofilm formation. The simultaneous repression of *ARO1* and aromatic amino acid uptake by carvacrol cause aromatic amino acid starvation. *ARO1* repression further eliminates co-enzyme Q precursors, leading to mitochondrial respiratory and metabolic dysfunction, disrupting adhesion, hyphal formation, and biofilm maturation. Verified mechanisms are indicated by solid lines, while proposed carvacrol mechanisms are indicated by dashed lines. Created in https://BioRender.com.

involved in both initial adhesion and adhesion maintenance in biofilms (33, 67, 68). We propose that carvacrol likely indirectly interferes with surface adhesin proteins Als1, Als3 and Hwp1 by compromising the plasma membrane.

Hyphal formation is a morphogenetic process that supports adhesion, invasion of host cells, and biofilm formation, contributing to *C. albicans* virulence and pathogenicity (69, 70). Strains with defective hyphae produce abnormal biofilms (71, 72), indicating its importance in providing strength and support for developing a mature biofilm (73). With exposure to carvacrol at sublethal levels, the repressed *ARO1* mutant takes on an irregular colony phenotype with little to no mycelial growth at the edge (Fig. 5), indicative of impaired morphological switching and hyphal formation (Fig. 6). However, the repressed *ARO2* and *ARO7* mutants under the same conditions had fewer morphological switching defects (Fig. 5b and d; Fig. S4). Repression of *ARO7* did not cause mitochondrial disfunction within biofilms (Fig. 7a and 8b) since the cell can still synthesize L-Trp, chorismate, and CoQ (Fig. 1). The unexpected difference in mycelial growth between the repressed *ARO1* and *ARO2* mutants when exposed to carvacrol at sublethal levels implies a potential Aro2 homolog or alternative pathway, requiring further investigation.

Our previous studies also report inhibition of *C. albicans* hyphal growth by carvacrol (54), but here, we show that carvacrol can revert pre-formed hyphae back to yeast for the repressed *ARO1* mutant (Fig. 6c and d), underscoring other possible roles for *ARO1* in the *C. albicans* life cycle and requiring further investigation. The hyphal to yeast reversion would interfere with biofilm formation and maturation, which explains how pre-formed biofilms are reduced by carvacrol in repressed *ARO1* (Fig. 8).

The conditional repression of *ARO1* increases sensitivity to carvacrol, as evidenced by reduced biofilm metabolic activity and biomass accumulation (Fig. 7 and 8). As a key component of the shikimate pathway, Aro1 is essential for cell viability, and its absence requires exogenous sources of aromatic amino acids. The shikimate pathway also produces co-enzyme Q (ubiquinone), required for respiratory electron transport in the inner mitochondrial membrane that plays a crucial role in oxidative phosphorylation (26). Mitochondrial respiration is required for morphological transition and biofilm formation in *C. albicans*, and thus its dysfunction can disrupt both processes (74), as supported by studies of mitochondrial inhibitors (75–78) (Fig. 9). The repression of *ARO1* should eliminate co-enzyme Q precursors (Fig. 3), resulting in mitochondrial respiratory and metabolic dysfunction, as evidenced by disruption of hyphal morphogenesis, biofilm metabolic activity and maturation (79) (Fig. 6 to 9).

It has been proposed that Gap1, Gap2, and Gap6 transporters may also serve as receptors for the activation of protein kinase A (PKA) in the Ras–cAMP–PKA pathway, which plays a key role morphogenesis and biofilm formation (29, 37). Carvacrol, given its impact on membrane potential (30), may reduce the activation of this pathway through the Gap permeases, thus also reducing the formation of robust and mature biofilm. This idea is supported by the significant reduction in metabolic activity and biomass accumulation in biofilms of the repressed *ARO1* strain pre-treated with carvacrol compared with repression alone (Fig. 7 and 8). Thus, a robust biofilm appears capable of excluding DOX, likely based on its extensive ECM (56).

## Conclusion

This study reveals that one of the ways carvacrol inhibits hyphal and biofilm formation in *Candida albicans* is by blocking amino acid uptake. The repression of *ARO1* eliminates co-enzyme Q precursors, resulting in mitochondrial respiratory and metabolic dysfunction, therefore disrupting hyphal growth and biofilm maturation. Thus, inhibitors of the Aro1 enzyme in combination with carvacrol are expected to attenuate *C. albicans* virulence by disrupting biofilm formation.

## ACKNOWLEDGMENTS

We thank Dr. Malcolm Whiteway (Department of Biology, Concordia University, Montreal, CA) for the kind gift of the strains used in this study. This research was conducted on the traditional territories of the Nêhiyawak, Anihsinapek, Nakoda, Dakota, and Lakota peoples and the homeland of the Métis/Michif Nation.

Fig 1 and 9 were created in BioRender. Molaeitabari, A. (2025) https://BioRender.com/k57r350 and https://BioRender.com/p62v865.

This study was supported by the Saskatchewan Health Research Foundation (SHRF 4769), Natural Science and Engineering Research Council (NSERC; 2018-06649; 2024-06684) and Canada Foundation for Innovation grants to TESD. The Faculty of Graduate Studies and Research at the University of Regina provided partial support to A.M.

Conceptualization, A.M. and T.E.S.D.; Methodology, A.M.; Validation, A.M.; Formal analysis, A.M.; Investigation, A.M.; Resources, T.E.S.D.; Data curation, A.M.; Writing - original draft, A.M.; Writing - review & editing, A.M. and T.E.S.D.; Supervision, T.E.S.D.; Project administration, T.E.S.D.; Funding acquisition, T.E.S.D. All authors have read and agreed to the published version of the manuscript.

## AUTHOR AFFILIATION

[1]Department of Chemistry and Biochemistry, University of Regina, Regina, Saskatchewan, Canada

## AUTHOR ORCIDs

Ali Molaeitabari http://orcid.org/0000-0002-6776-5214
Tanya E. S. Dahms http://orcid.org/0000-0002-8378-7480

## FUNDING

| Funder | Grant(s) | Author(s) |
| --- | --- | --- |
| Canadian Government \| Natural Sciences and Engineering Research Council of Canada (NSERC) | 2024-06684 | Tanya E. S. Dahms |
| Saskatchewan Health Research Foundation (SHRF) | 4769 | Tanya E. S. Dahms |
| Canadian Government \| Natural Sciences and Engineering Research Council of Canada (NSERC) | 2018-06649 | Tanya E. S. Dahms |

## AUTHOR CONTRIBUTIONS

Ali Molaeitabari, Conceptualization, Data curation, Formal analysis, Investigation, Methodology, Visualization, Writing – original draft | Tanya E. S. Dahms, Conceptualization, Funding acquisition, Project administration, Resources, Supervision, Writing – review and editing

## ADDITIONAL FILES

The following material is available online.

### Supplemental Material

**Figures S1 to S4 (Spectrum02754-24-s0001.pdf).** Growth curves, gene essentiality screen, full PABA rescue experiments, carvacrol impact on mycelial growth.

### Open Peer Review

**PEER REVIEW HISTORY (review-history.pdf).** An accounting of the reviewer comments and feedback.

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
