## [Reviewer comments · Microbiology Spectrum]

Microbiology Spectrum

Blocking the shikimate pathway amplifies the impact of carvacrol on biofilm formation in *Candida albicans*

Ali Molaeitabari and Tanya Dahms

Corresponding Author(s): Tanya Dahms, University of Regina Department of Chemistry and Biochemistry

Review Timeline:

Submission Date:	November 5, 2024
Editorial Decision:	November 25, 2024
Revision Received:	December 4, 2024
Accepted:	December 14, 2024

Editor: Renato Kovacs

Reviewer(s): Disclosure of reviewer identity is with reference to reviewer comments included in decision letter(s). The following individuals involved in review of your submission have agreed to reveal their identity: Alireza Khodavandi (Reviewer #1)

Transaction Report:

DOI: <https://doi.org/10.1128/spectrum.02754-24>

Re: Spectrum02754-24 (Blocking the shikimate pathway amplifies the impact of carvacrol on biofilm formation in *Candida albicans*)

Dear Dr. Tanya Elizabeth Susan Dahms:

Thank you for the privilege of reviewing your work. Below you will find my comments, instructions from the Spectrum editorial office, and the reviewer comments.

Revision Guidelines

Sincerely,
Renato Kovacs
Editor
Microbiology Spectrum

Reviewer #1 (Comments for the Author):

The manuscript by Molaeitabari and Dahms is "Blocking the shikimate pathway amplifies the impact of carvacrol on biofilm formation in *Candida albicans*" is interesting relatively and could be sufficient for publication in Microbiology Spectrum but not in this format and needs minor corrections. Indeed, the methods are enough. Some phrases were not clear and need to rewrite. However, author is kindly requested to address the following issues.

Title: OK

Abstract: Abstract is poor section and need to be extended by using more significant results. I recommend to use graphical

abstract.

Introduction:

- 1- Page 5 line 110 Put in quotation marks instead of parentheses.
- 2- The objectives of this research are not well stated and need more clarification.
- 3- Moreover, Innovation is not obvious. Please clarify

Materials and methods:

- 1- All methods need to be addressed.
- 2- Why you did not use sub-MIC such as MIC50% or MIC90%?
- 3- Based on CLSI documents, OD for yeast must be 530 nm while 600 nm is suitable for bacteria.

Overall, I believe the antifungal methods were enough and could improve in this part.

Results and Discussion: In my opinion results of this study are sufficient but, I recommend to clarify and describe more. On the other hand, I recommend to add and use some articles and improve discussion part as bellows:

- Probable Molecular Targeting of the Inhibitory Effect of Carvacrol-Loaded Bovine Serum Albumin Nanoparticles on Human Breast Adenocarcinoma Cells (MCF-7 Cells)
- Study of the antifungal potential of carvacrol on growth inhibition of *Candida krusei* in a systemic candidiasis
- Microbiological and Histological Characteristics of Interactions Between Carvacrol and Fluconazole in a Systemic Candidiasis Animal Model

Reviewer #2 (Comments for the Author):

Dear Authors

I carefully reviewed your study titled "Blocking the shikimate pathway amplifies the impact of carvacrol on biofilm formation in *Candida albicans*" and found it really interesting. This study, in which you examine the effects of carvacrol on *Candida albicans* biofilm formation and the potential of shikimate pathway genes, fills an important gap in the field.

Your paper has a strong methodology and your experimental design is very robust. Your detailed discussion of the effects of carvacrol on biofilm formation and cellular structures is a valuable contribution to the literature. Also, your emphasis in the discussion section on how the findings can be used in a clinical context is very important and instructive.

I have only a few minor revisions to suggest for the acceptance of the study. I believe that once these revisions are made, your study will become much stronger.

Best regards.

Introduction Section

The introduction provides a good overview of the pathogenic properties of *Candida albicans* and the potential for antifungal effects of carvacrol. The mention of the importance of targets such as the shikimate pathway in antifungal therapies frames the context of the study well.

Recommendations:

- Lines 40-44: The clinical significance of candidiasis could be supported by more specific statements. This would strengthen the clinical context of the study.
- Line 55: Superficial mention of the impact of *Candida albicans* biofilm formation on infections. A sentence explaining that biofilm formation increases resistance to antifungal treatments could be added.

Materials and Methods Section

The methods used provide a strong basis for relevance, comprehensiveness and reproducibility.

Recommendations:

- Line 139-140: The determination of the mid-logarithmic phase and the time intervals covered by this phase is critical information for experimental reproducibility. It would be appropriate to indicate which hour was determined as the mid-logarithmic phase in the study. If this time interval was found experimentally, this should be explained or if it was decided based on the literature, the relevant article should be cited.
- Line 140: It can be indicated how many ml of culture was sub-cultured by adding how many ml of culture to how many ml of fresh SC or RPMI.

Results Section

The findings clearly reveal the effects of carvacrol on both planktonic cells and biofilm formation. However, Figure 5 a shows images of both $\frac{1}{4}$ and $\frac{1}{2}$ MIC study in the continuous exposure to carvacrol (6 days) group. In Figure 5 b, only the images and table of the $\frac{1}{2}$ MIC study in the absence of carvacrol group after 4 hours pretreatment are given. Why $\frac{1}{4}$ MIC results are also not shown in this section. It would be appropriate to present $\frac{1}{4}$ MIC results in Figure 5b.

Discussion Section

Lines 438-440: In the Discussion it is stated that carvacrol inhibits hyphal morphogenesis in the suppressed ARO1 mutant. However, it is not discussed why this mechanism is more pronounced compared to other mutants (ARO2 and ARO7). More support can be added from the literature on how ARO1 affects cell wall proteins differently from other enzymes or its relationship with mitochondrial functions.

Response to Reviewers

Changes to the manuscript appear in blue font.

Reviewer #1 (Comments for the Author):

The manuscript by Molaeitabari and Dahms is "Blocking the shikimate pathway amplifies the impact of carvacrol on biofilm formation in *Candida albicans*" is interesting relatively and could be sufficient for publication in Microbiology Spectrum but not in this format and needs minor corrections. Indeed, the methods are enough. Some phrases were not clear and need to rewrite.

However, author is kindly requested to address the following issues.

We appreciate your time and kind review.

Title:

OK

Abstract:

Abstract is poor section and need to be extended by using more significant results. I recommend to use graphical abstract.

We thank you for this comment and have revised the abstract accordingly (lines 23-26). This journal does not use the graphical abstract and the data has been summarized in Figure 9.

Introduction:

1- Page 5 line 110 Put in quotation marks instead of parentheses.

We appreciate your perspective, but convention dictates the use of parentheses for acronyms related to proteins and genes.

2- The objectives of this research are not well stated and need more clarification.

Thank you, you are correct. We have revised this (lines 127-129).

3- Moreover, Innovation is not obvious. Please clarify

Thank you for pointing out this oversight that we have revised (lines 129-131).

Materials and methods:

1- All methods need to be addressed.

We appreciate this comment and have included more details wherever possible (lines 156-161), and most of the methods are already heavily referenced. If there are still missing details, could the reviewer please expound?

2- Why did you not use sub-MIC such as MIC50% or MIC90%?

We have used the 1/4, 1/2 and full MIC designations in our previous papers (see below), and so we do this for consistency.

Shahina Z., Molaeitabari A., Sultana T., Dahms T.E.S.* (2022). Cinnamon leaf and clove essential oils are potent inhibitors of *Candida albicans* virulence traits. *Microorganisms*; 10, 1989. <https://doi.org/10.3390/microorganisms10101989>

Shahina Z., Ndlovu E., Persaud O., Sultana T., Dahms T.E.S.* (2022). *Candida albicans* reactive oxygen species (ROS)-dependent lethality and ROS-independent hyphal and biofilm inhibition by eugenol and citral. *Microbiology Spectrum*;10(6):e0318322. doi: 10.1128/spectrum.03183-22. PMID: 36394350; PMCID: PMC9769929.

Acuna E., Ndlovu E., Molaeitabari A., Shahina Z.*, Dahms T. E. S.* (2023) Carvacrol-Induced Vacuole Dysfunction and Morphological Consequences in *Nakaseomyces glabratus* and *Candida albicans*. *Microorganisms* 2023, 11, 2915. <https://doi.org/10.3390/microorganisms11122915>

3- Based on CLSI documents, OD for yeast must be 530 nm while 600 nm is suitable for bacteria.

We thank you for your comment regarding the use of OD530 for yeast, as recommended by the CLSI guidelines for antifungal susceptibility testing, but optical density readings at 600 nm reflect optical scattering from organisms, whether they are bacteria or yeast.

Although OD530 is mentioned in the CLSI guidelines, using OD600 is not strictly prohibited by the CLSI and OD600 is extensively used in similar yeast studies (see below) for growth curves, antifungal susceptibility testing, and other biological and microbiological assays. The value is an established standard that ensures reproducibility and facilitates comparisons across studies.

1. Shahina Z, Homsy R Al, Price JDW, Whiteway M, Sultana T, Dahms TES. Rosemary essential oil and its components 1,8-cineole and α -pinene induce ROS-dependent lethality and ROS-independent virulence inhibition in *Candida albicans* [Internet]. Vol. 17, PLoS ONE. 2022. 1–31 p. Available from: <http://dx.doi.org/10.1371/journal.pone.0277097>

2. Roemer T, Jiang B, Davison J, Ketela T, Veillette K, Breton A, et al. Large-scale essential gene identification in *Candida albicans* and applications to antifungal drug discovery. *Mol Microbiol.* 2003;50(1):167–81. <https://doi.org/10.1046/j.1365-2958.2003.03697.x>

3. Omeara TR, Veri AO, Ketela T, Jiang B, Roemer T, Cowen LE. Global analysis of fungal morphology exposes mechanisms of host cell escape. *Nat Commun.* 2015;6:1–10. <https://doi.org/10.1038/ncomms7741>

4. Segal ES, Gritsenko V, Levitan A, Yadav B, Dror N, Steenwyk JL, et al. Gene essentiality analyzed by in vivo transposon mutagenesis and machine learning in a stable haploid isolate of *Candida albicans*. *MBio.* 2018;9(5):e02048-18. <https://doi.org/10.1128/mbio.02048-18>

5. Shahina Z, Ndlovu E, Persaud O, Sultana T, Dahms TES. Candida albicans reactive oxygen species (ROS)-dependent lethality and ROS-independent hyphal and biofilm inhibition by eugenol and citral. *Microbiol Spectr.* 2022;10(6):e03183-22.

<https://doi.org/10.1128/spectrum.03183-22>

6. Darvishi E, Omid M, Bushehri AA, Golshani A, Smith ML. The antifungal eugenol perturbs dual aromatic and branched-chain amino acid permeases in the cytoplasmic membrane of yeast. *PLoS One.* 2013/11/10. 2013;8(10):e76028.

<https://doi.org/10.1371/journal.pone.0076028>

7. Rao A, Zhang Y, Muend S, Rao R. Mechanism of antifungal activity of terpenoid phenols resembles calcium stress and inhibition of the TOR pathway. *Antimicrob*

<https://doi.org/10.1128/aac.01050-10>

Overall, I believe the antifungal methods were enough and could improve in this part.

We do not understand this comment.

Results and Discussion:

In my opinion results of this study are sufficient but, I recommend to clarify and describe more.

Thank you, but the reviewer would have to be more specific in this regard. In the results section we fully describe our data, including that presented within the supplemental section, and in the discussion section we contextualize our data according to the current literature.

If the reviewer has some specific suggestions, we will be happy to include additional information.

On the other hand, I recommend to add and use some articles and improve discussion part as bellows:

-Probable Molecular Targeting of the Inhibitory Effect of Carvacrol-Loaded Bovine Serum Albumin Nanoparticles on Human Breast Adenocarcinoma Cells (MCF-7 Cells)

-Study of the antifungal potential of carvacrol on growth inhibition of *Candida krusei* in a systemic candidiasis

-Microbiological and Histological Characteristics of Interactions Between Carvacrol and Fluconazole in a Systemic Candidiasis Animal Model

Thank you for these suggestions, and we have added the second and third articles an additional text to our Introduction (lines 107-112). However, the first article focuses on the effect of carvacrol on breast cancer, and so was not relevant to this research. Since the second and third articles deal with the impact of carvacrol on systemic candidiasis with

non-*C. albicans* species in mouse model, they were not relevant to the data presented and therefore not included in our discussion.

Reviewer #2 (Comments for the Author):

Dear Authors

I carefully reviewed your study titled "Blocking the shikimate pathway amplifies the impact of carvacrol on biofilm formation in *Candida albicans*" and found it really interesting. This study, in which you examine the effects of carvacrol on *Candida albicans* biofilm formation and the potential of shikimate pathway genes, fills an important gap in the field.

Your paper has a strong methodology, and your experimental design is very robust. Your detailed discussion of the effects of carvacrol on biofilm formation and cellular structures is a valuable contribution to the literature. Also, your emphasis in the discussion section on how the findings can be used in a clinical context is very important and instructive.

I have only a few minor revisions to suggest for the acceptance of the study. I believe that once these revisions are made, your study will become much stronger.

Best regards.

We appreciate your time and careful review.

Introduction Section

The introduction provides a good overview of the pathogenic properties of *Candida albicans* and the potential for antifungal effects of carvacrol. The mention of the importance of targets such as the shikimate pathway in antifungal therapies frames the context of the study well.

Recommendations:

- Lines 40-44: The clinical significance of candidiasis could be supported by more specific statements. This would strengthen the clinical context of the study.

Thank you for these suggestions, and we have added a paragraph that outlines the clinical significance of candidiasis (lines 43-50).

- Line 55: Superficial mention of the impact of *Candida albicans* biofilm formation on infections. A sentence explaining that biofilm formation increases resistance to antifungal treatments could be added.

We thank you for your suggestion, but we have already included this in the introduction (line 62).

Materials and Methods Section

The methods used provide a strong basis for relevance, comprehensiveness and reproducibility.

Recommendations:

- Line 139-140: The determination of the mid-logarithmic phase and the time intervals covered by this phase is critical information for experimental reproducibility. It would be appropriate to indicate which hour was determined as the mid-logarithmic phase in the study. If this time interval was found experimentally, this should be explained or if it was decided based on the literature, the relevant article should be cited.

Thank you for pointing out this omission. We have added this additional detail to the material methods (lines 156-161). We have also included growth curves in the revised supplemental material (Figure S1).

- Line 140: It can be indicated how many ml of culture was sub-cultured by adding how many ml of culture to how many ml of fresh SC or RPMI.

Thank you for pointing out this omission. We have added this additional detail to the material methods (line 160-161).

Results Section

The findings clearly reveal the effects of carvacrol on both planktonic cells and biofilm formation. However, Figure 5 a shows images of both $\frac{1}{4}$ and $\frac{1}{2}$ MIC study in the continuous exposure to carvacrol (6 days) group. In Figure 5 b, only the images and table of the $\frac{1}{2}$ MIC study in the absence of carvacrol group after 4 hours pretreatment are given. Why $\frac{1}{4}$ MIC results are also not shown in this section. It would be appropriate to present $\frac{1}{4}$ MIC results in Figure 5b.

The assay preliminary to generating Figure 5b showed no difference between control and $\frac{1}{4}$ MIC, nor did we see a difference between control and $\frac{1}{4}$ MIC for the mycelial assay, which is consistent with our MIC results (ARO1 control and $\frac{1}{4}$ MIC (31.25 μ g/mL)) (Figure 2), so we chose courser MIC treatments (0, $\frac{1}{2}$, full) for this more complex and laborious assay shown in Figure 5b.

For the mycelial growth inhibition assay, the cells were directly exposed to carvacrol or DOX in solid medium over the 6 days, which is non-ideal for *C. albicans* growth. Thus,ed more fractional concentrations were used to ensure growth.

Discussion Section

Lines 438-440: In the Discussion it is stated that carvacrol inhibits hyphal morphogenesis in the suppressed ARO1 mutant. However, it is not discussed why this mechanism is more pronounced compared to other mutants (ARO2 and ARO7). More support can be added from the literature on how ARO1 affects cell wall proteins differently from other enzymes or its relationship with mitochondrial functions.

Thank you for this suggestion, we now have provided a comparison between the morphological switching of *ARO1* mutants and the other mutants and discussed possible reasons (line 460-466).

Re: Spectrum02754-24R1 (Blocking the shikimate pathway amplifies the impact of carvacrol on biofilm formation in *Candida albicans*)

Dear Dr. Tanya Elizabeth Susan Dahms:

Your manuscript has been accepted, and I am forwarding it to the ASM production staff for publication. Your paper will first be checked to make sure all elements meet the technical requirements. ASM staff will contact you if anything needs to be revised before copyediting and production can begin. Otherwise, you will be notified when your proofs are ready to be viewed.

Sincerely,
Renato Kovacs
Editor
Microbiology Spectrum